# Gaussian Splatting Under Attack: Investigating Adversarial Noise in 3D Objects

**Abdurrahman Zeybey**     **Mehmet Ergezer**\*     **Tommy Nguyen**
School of Computing and Data Science
Wentworth Institute of Technology
Boston, MA
{zeybeya,ergezerm,nguyent68}@wit.edu

## Abstract

3D Gaussian Splatting has advanced radiance field reconstruction, enabling high-quality view synthesis and fast rendering in 3D modeling. While adversarial attacks on object detection models are well-studied for 2D images, their impact on 3D models remains underexplored. This work introduces the Masked Iterative Fast Gradient Sign Method (M-IFGSM), designed to generate adversarial noise targeting the CLIP vision-language model. M-IFGSM specifically alters the object of interest by focusing perturbations on masked regions, degrading the performance of CLIP's zero-shot object detection capability when applied to 3D models. Using eight objects from the Common Objects 3D (CO3D) dataset, we demonstrate that our method effectively reduces the accuracy and confidence of the model, with adversarial noise being nearly imperceptible to human observers. The top-1 accuracy in original model renders drops from 95.4% to 12.5% for train images and from 91.2% to 35.4% for test images, with confidence levels reflecting this shift from true classification to misclassification, underscoring the risks of adversarial attacks on 3D models in applications such as autonomous driving, robotics, and surveillance. The significance of this research lies in its potential to expose vulnerabilities in modern 3D vision models, including radiance fields, prompting the development of more robust defenses and security measures in critical real-world applications.

## 1   Introduction

Computer vision has rapidly advanced with the development of large-scale datasets and vision-language models such as CLIP Li u. a. (2022). These advancements have expanded the application of computer vision algorithms into areas critical to modern life, such as humanoid robots, autonomous driving, and surveillance systems Kim u. a. (2024)Zhou u. a. (2024). Ensuring the accuracy and robustness of these systems is crucial, as errors could lead to severe consequences, from accidents in autonomous vehicles to failures in surveillance systems.

While adversarial attacks on 2D vision models have been widely studied Goodfellow u. a. (2015); Szegedy u. a. (2014), their impact on 3D models remains underexplored Li u. a. (2024); Song u. a. (2024). This gap is significant, as 3D models are increasingly used in real-world applications, including robotics, augmented reality, and autonomous systemsZhu u. a. (2023); Zhao u. a. (2019). Recent progress in 3D model rendering techniques, such as 3D Gaussian Splatting (3DGS) Kerbl u. a. (2023), has enabled high-quality view synthesis and efficient rendering in 3D modeling, presenting new opportunities, and vulnerabilities, for adversarial attacksZeng u. a. (2019). However, there is

---

\*Dr. Ergezer holds concurrent appointments as an Associate Professor at Wentworth Institute of Technology and as an Amazon Visiting Academic. This paper describes work performed at Wentworth Institute of Technology and is not associated with Amazon.

38th Conference on Neural Information Processing Systems (NeurIPS 2024).

limited research on the robustness of 3D models against such attacks, especially in vision-language contexts Zou u. a. (2023).

In this work, we introduce the **Masked Iterative Fast Gradient Sign Method (M-IFGSM)**, a novel adversarial attack designed to degrade the performance of the CLIP vision-language model in 3D object detection. Unlike traditional 2D adversarial attacks Madry u. a. (2018), M-IFGSM focuses perturbations on masked regions of 3D objects, creating adversarial noise that remains nearly imperceptible to humans while significantly degrading model accuracy. This masked approach allows for a more targeted attack, where only specific regions of the 3D object are perturbed, making detection of the adversarial noise more challenging.

We validate our approach through extensive experiments using the CO3D dataset Reizenstein u. a. (2021), demonstrating that M-IFGSM can substantially lower CLIP's top-1 accuracy from 95.4% to 12.5% on training images, and from 91.2% to 35.4% on test images. These results illustrate the potential risks posed by adversarial attacks on 3D models, which are becoming increasingly prevalent in safety-critical applications such as autonomous driving, robotics, and surveillance. By exposing vulnerabilities in vision-language models when applied to 3D objects, this research underscores the pressing need to develop robust defense mechanisms in these high-stakes domains.

Our contributions are threefold: *i)* First, we propose the M-IFGSM method, which introduces a novel approach to adversarial attacks in 3D models within a vision-language context. *ii)* Second, we leverage 3D Gaussian Splatting to generate adversarial perturbations that are localized to masked regions, providing a fine-grained method to compromise model performance. *iii)* Finally, we emphasize the real-world implications of such attacks, underscoring the urgent need to develop robust defenses for systems relying on 3D object detection in critical applications. Our approach significantly reduces CLIP's accuracy in 3D object detection with minimal, almost imperceptible noise, resulting in a substantial drop in top-1 accuracy across both train and test datasets.

## 2 Method

In this section, we introduce our proposed method, the **Masked Iterative Fast Gradient Sign Method (M-IFGSM)**, designed to generate adversarial perturbations specifically targeting 3D models in a vision-language context. While most adversarial methods perturb the entire image uniformly, this can lead to suboptimal attacks and degrade downstream tasks, such as 3D reconstruction. Our approach enhances traditional adversarial attack methods by focusing perturbations solely on the object of interest within input images. We detail the components of our pipeline, including segmentation using the Segment Anything Model (SAM), adversarial perturbation generation with M-IFGSM, and 3D model reconstruction using Gaussian Splatting.

Our method consists of three main stages: **i) Mask Generation:** Utilizing the Segment Anything Model (SAM) to extract object masks from input images. **ii) Adversarial Perturbation Generation:** Applying M-IFGSM to generate noise focused on masked regions. **iii) 3D Model Reconstruction:** Building 3D models from adversarial images using Gaussian Splatting. Figure 1 illustrates the overall pipeline.

### 2.1 Mask Generation with SAM

We employ the Segment Anything Model (SAM) Kirillov u. a. (2023) to generate accurate segmentation masks for the target objects within the input images. Utilizing the pre-trained 'sam-vit-h-4b899' checkpoint, SAM provides high-quality masks without the need for additional training. The mask generation process outputs the segmentation mask $\mathbf{M}_i$, bounding boxes, and quality metrics.

### 2.2 Adversarial Perturbation Generation with M-IFGSM

Traditional FGSM applies uniform perturbations across the entire image, often leading to undesirable artifacts in 3D reconstructions due to background noise. To address this, we propose M-IFGSM, which enhances FGSM by concentrating perturbations solely on the segmented object regions. This targeted approach improves attack effectiveness and preserves background integrity.

The untargeted attack is formulated as:

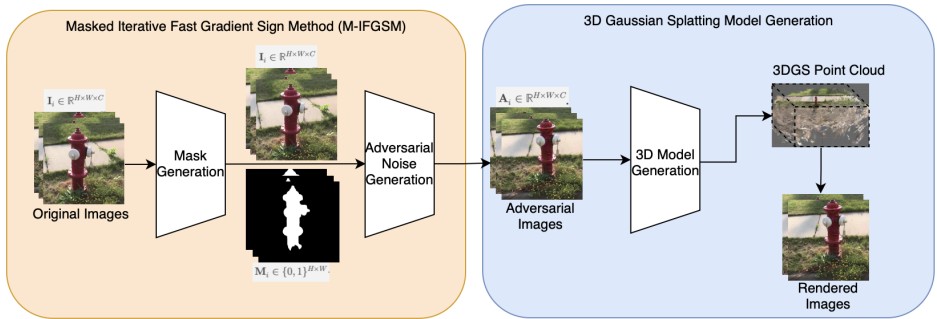

Figure 1: Two-stage pipeline for generating adversarial 3D models using the proposed Masked Iterative Fast Gradient Sign Method (M-IFGSM) and 3D Gaussian Splatting. In the first stage (left), original images $\mathbf{I}_i$ are processed to generate a mask $\mathbf{M}_i$, which specifies the regions where adversarial noise will be applied. Using this mask, adversarial perturbations are crafted to produce adversarial images $\mathbf{A}_i$, which contain noise designed to mislead detection systems. In the second stage (right), these adversarial images are used to create a 3D model through 3D Gaussian Splatting, resulting in a 3DGS point cloud that captures the adversarial characteristics in a 3D format. Finally, the 3D model is rendered to produce images that retain the adversarial properties, allowing for an evaluation of adversarial noise effects in 3D object detection.

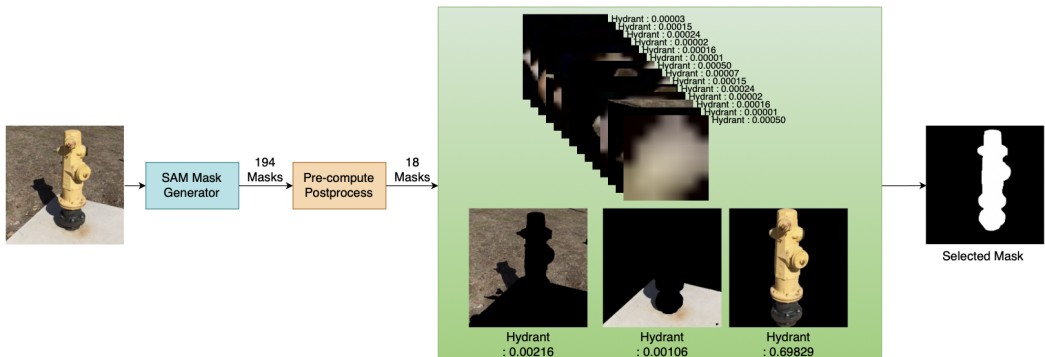

Figure 2: Mask generation pipeline

$$X_{N+1}^{\mathrm{adv}} = \mathrm{Clip}_{\min,\max}\left\{X_{\mathrm{inv}} + \mathbf{M} \odot \left(X_N^{\mathrm{adv}} + \epsilon \cdot \mathrm{sign}\left(\nabla_{X_N} J(X_N^{\mathrm{adv}}, y_{\mathrm{true}})\right)\right)\right\} \quad (1)$$

For targeted attacks:

$$X_{N+1}^{\mathrm{adv}} = \mathrm{Clip}_{\min,\max}\left\{X_{\mathrm{inv}} + \mathbf{M} \odot \left(X_N^{\mathrm{adv}} - \epsilon \cdot \mathrm{sign}\left(\nabla_{X_N} J(X_N^{\mathrm{adv}}, y_{\mathrm{target}})\right)\right)\right\} \quad (2)$$

where $X_N^{\mathrm{adv}}$ is the adversarial image at iteration $N$; $\mathbf{M}$ is the segmentation mask; $X_{\mathrm{inv}}$ is the inverse image (background); $\epsilon$ controls the perturbation magnitude; $J$ represents the loss function (cross-entropy loss), $\mathrm{Clip}_{\min,\max}$ keeps the values within the RGB range of 0 to 255 and $\odot$ is the element-wise multiplication operation.

Algorithm 1 details the iterative process for generating adversarial images. In our experiments, we set the loss threshold $\tau$ to 20.00 for early stopping.

Our pipeline is designed with flexibility in mind, allowing it to generate adversarial noise for a variety of object classification models, not solely the CLIP model. Our methodology leverages a Masked Iterative Fast Gradient Sign Method (M-IFGSM) that can target any differentiable model f to compute gradients and generate adversarial perturbations. This adaptability allows our pipeline to extend

to various vision models, providing a framework that can be tested on other object classification systems.

---

**Algorithm 1** Masked Iterative FGSM Adversarial Attack

---

**Input:** Image $X$, Segmentation Mask $\mathbf{M}$, Model $f$, True Label $y_{\text{true}}$
**Parameters:** Learning Rate $\epsilon$, Number of Iterations $N$
**Output:** Adversarial Image $X^{\text{adv}}$

1: Compute inverse image: $X_{\text{inv}} = X \odot (1 - \mathbf{M})$
2: Initialize adversarial image: $X_0^{\text{adv}} = X$
3: **for** $n = 0$ to $N - 1$ **do**
4:      Enable gradient computation for $X_n^{\text{adv}}$
5:      Compute loss: $J = \text{CrossEntropy}(f(X_n^{\text{adv}}), y_{\text{true}})$
6:      Backpropagate to obtain gradients: $\nabla_X J$
7:      Update adversarial image:

$$X_{n+1}^{\text{adv}} = \text{Clip}_{\min,\max} \left\{ X_{\text{inv}} + \mathbf{M} \odot \left( X_n^{\text{adv}} + \epsilon \cdot \text{sign}(\nabla_X J) \right) \right\}$$

8:      Early stopping if $probs[y_{\text{true}}] = 0$ and $J > \tau$
9: **end for**
10: Return $X^{\text{adv}} = X_N^{\text{adv}}$

---

### 2.3 3D Model Reconstruction

After generating adversarial images with M-IFGSM, we reconstruct 3D models using the Gaussian Splatting method Kerbl u. a. (2023). We use an 85-15% train-test split, with 35 images for training and six for testing.

The reconstruction process involves four stages. **i) Initialization:** Using a sparse point cloud from Structure from Motion Schonberger und Frahm (2016) to initialize 3D Gaussians. **ii) Optimization:** Adjusting the Gaussians' parameters (position, covariance, opacity, color) via gradient descent to match the adversarial images. **iii) Adaptive Density Control:** Refining the model by adding or splitting Gaussians in under- or over-reconstructed areas. **iiii) Rendering:** Employing a tile-based rasterizer for efficient real-time visualization.

This process allows us to assess how adversarial perturbations affect 3D model accuracy and the performance of the CLIP model.

### 2.4 Experimental Setup

We refer to the attacked model as the victim. We target the CLIP ViT-B/16 model Dosovitskiy u. a. (2021), which combines vision and language understanding. The model divides input images into $16 \times 16$ patches and processes them using transformer layers.

For our dataset, we employed the Common Objects in 3D (CO3D) dataset Reizenstein u. a. (2021), selecting one dataset for each of the eight object classes. Each dataset contains approximately 200 images from different angles. We reduce the number to 41 images per class by selecting every fifth image and resizing them to $224 \times 224$ pixels to meet the input requirements of CLIP. Our experiments are conducted on a system with dual NVIDIA RTX 3090 GPUs.

## 3 Results

In this section, we present and analyze the outcomes of our experiments conducted to evaluate the efficacy of M-IFGSM for generating adversarial noise in the context of 3D Gaussian Splatting (3DGS). Our primary objective was to assess the impact of adversarial attacks on the classification performance of the object detection model when subjected to adversarially perturbed 3D models. First, we discuss the results of perturbed images generated by the M-IFGSM method, followed by the rendered images generated by the 3DGS technique.

### 3.1 Perturbed Images Results

Table 1 presents the classification confidence and accuracy for perturbed images generated using M-IFGSM. The "Original Images" column contains the confidence and top-1/top-5 accuracy for unmodified images, serving as baseline performance. The "Adversarial Images" column shows the same statistics for perturbed images, highlighting the degradation in the model's confidence and accuracy due to the adversarial perturbations.

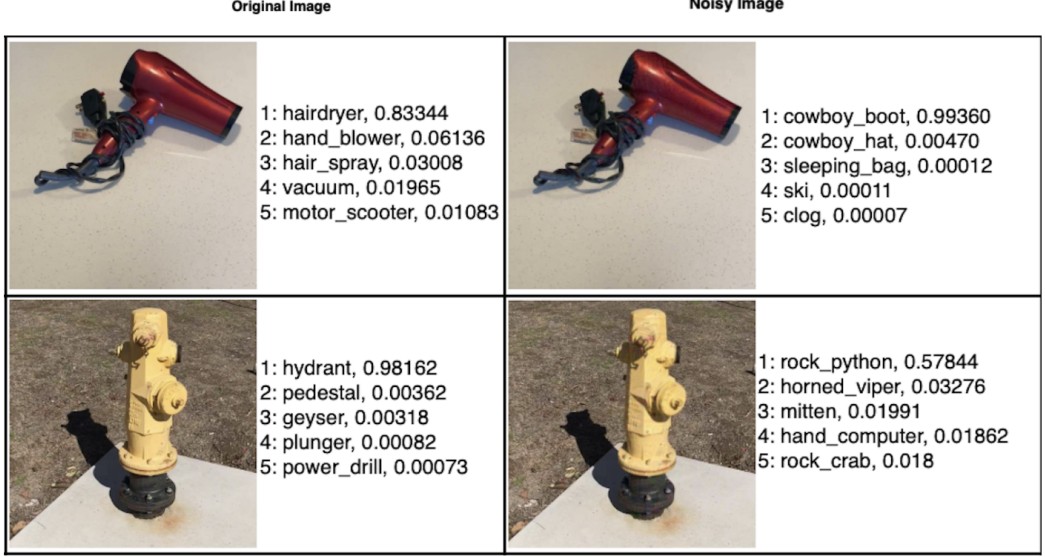

Figure 3: Adversarial noise effects on object detection of the hairdryer and hydrant objects. Sample images of the original image results (left top and bottom) and their attacked counterparts (right top and bottom).

Figures 3 provide visual confirmation of the adversarial noise's effect, as the attacked objects show clear degradation in classification confidence when compared to their original counterparts.

Table 1: Average CLIP-ViT-B/16 classification results for input images and adversarial images generated with M-IFGSM. $Confidence^1$ refers to the average confidence assigned to the correct class for original images, while $Confidence^2$ is the confidence in incorrect predictions for adversarial images. "Top1" and "Top5" represent the occurrence of the true label in the top-1 and top-5 predictions, respectively.

| Object | Original Images | | | Adversarial Images | | |
|---|---|---|---|---|---|---|
| | $Confidence^1$ | Top1 | Top5 | $Confidence^2$ | Top1 | Top5 |
| Apple | 0.743 | 1.000 | 1.000 | 0.998 | 0.000 | 0.000 |
| Cake | 0.801 | 1.000 | 1.000 | 0.992 | 0.000 | 0.000 |
| Couch | 0.843 | 1.000 | 1.000 | 0.666 | 0.122 | 0.463 |
| Hairdryer | 0.705 | 0.854 | 1.000 | 0.909 | 0.000 | 0.000 |
| Hydrant | 0.971 | 1.000 | 1.000 | 0.656 | 0.000 | 0.000 |
| Motorcycle | 0.505 | 0.917 | 1.000 | 0.999 | 0.000 | 0.000 |
| Mouse | 0.579 | 0.821 | 0.974 | 0.887 | 0.000 | 0.051 |
| Suitcase | 0.689 | 1.000 | 1.000 | 0.934 | 0.000 | 0.000 |
| Average | 0.730 | 0.949 | 0.996 | 0.880 | 0.021 | 0.064 |

The average results show that for original images, the true label appears in the top-1 predictions 94.9% of the time and in the top-5 predictions 99.6% of the time. After the application of M-IFGSM, these figures drop significantly, with top-1 accuracy falling to just 2.1%, and top-5 accuracy to 6.4%. The model's misclassification confidence increases dramatically for adversarially perturbed images, showcasing the method's effectiveness.

The Adversarial Images column in Table 1 provides the same statistics for the perturbed images generated using the M-IFGSM method. These values reflect the impact of the adversarial perturbations on the classification performance, highlighting the effectiveness of the M-IFGSM method in degrading the model's confidence and accuracy. By comparing the results between the original and adversarial images, we can quantitatively assess the vulnerability of the object detection system to the adversarial attack.

## 3.2 Rendered Images Results

The additional results on the rendered images further highlight the impact of adversarial perturbations on 3D models. In Table 2, we present the results of the classification confidence and accuracy of rendered images from the 3D Gaussian Splatting (3DGS) models. The original 3DGS model was created with a clean image dataset, while the adversarial 3DGS model was created with perturbed images. At this point, we split the images into 85% train and 15% test sets, meaning 35 out of 41 images were used in the 3D modeling process, while 6 out of 41 images' camera positions were used as test renders after the models were created.

Table 2: Average CLIP-ViT-B/16 classification results for original 3DGS model renders and adversarial 3DGS model renders. $Confidence^1$ refers to the average confidence assigned to the correct class for original model renders, while $Confidence^2$ shows the confidence in incorrect predictions for adversarial model renders. "Top1" and "Top5" represent the occurrence of the true label in the top-1 and top-5 predictions, respectively.

| Object | Original Model Renders | | | Adversarial Model Renders | | |
|---|---|---|---|---|---|---|
| | $Confidence^1$ | Top1 | Top5 | $Confidence^2$ | Top1 | Top5 |
| Apple (Train) | 0.743 | 1.000 | 1.000 | 0.977 | 0.000 | 0.485 |
| Apple (Test) | 0.746 | 1.000 | 1.000 | 0.566 | 0.333 | 1.000 |
| Cake (Train) | 0.804 | 1.000 | 1.000 | 0.686 | 0.114 | 0.342 |
| Cake (Test) | 0.777 | 1.000 | 1.000 | 0.498 | 0.333 | 0.833 |
| Couch (Train) | 0.844 | 1.000 | 1.000 | 0.509 | 0.800 | 1.000 |
| Couch (Test) | 0.836 | 1.000 | 1.000 | 0.529 | 0.833 | 1.000 |
| Hairdryer (Train) | 0.700 | 0.857 | 1.000 | 0.372 | 0.000 | 0.114 |
| Hairdryer (Test) | 0.739 | 0.833 | 1.000 | 0.185 | 0.166 | 0.500 |
| Hydrant (Train) | 0.970 | 1.000 | 1.000 | 0.374 | 0.085 | 0.200 |
| Hydrant (Test) | 0.975 | 1.000 | 1.000 | 0.764 | 0.833 | 1.000 |
| Motorcycle (Train) | 0.508 | 0.952 | 1.000 | 0.993 | 0.000 | 0.381 |
| Motorcycle (Test) | 0.486 | 0.666 | 1.000 | 0.423 | 0.333 | 1.000 |
| Mouse (Train) | 0.573 | 0.823 | 0.970 | 0.580 | 0.000 | 0.088 |
| Mouse (Test) | 0.628 | 0.800 | 1.000 | 0.226 | 0.0 | 0.400 |
| Suitcase (Train) | 0.691 | 1.000 | 1.000 | 0.690 | 0.000 | 0.200 |
| Suitcase (Test) | 0.670 | 1.000 | 1.000 | 0.468 | 0.000 | 0.666 |
| Average (Train) | 0.729 | 0.954 | 0.996 | 0.648 | 0.125 | 0.351 |
| Average (Test) | 0.732 | 0.912 | 1.000 | 0.457 | 0.354 | 0.800 |

The notable case of the couch dataset, where the segmentation process only perturbs one of two couches in the image, underscores a limitation in current masking techniques when dealing with multiple instances of the same class in a scene.

Figure 4 particularly emphasizes the model's struggle when confronted with novel views of the object, where the adversarial perturbation has a diminished impact due to the lack of adaptation for unseen camera positions.

Results indicate that renders from training camera locations successfully transfer the adversarial noise into 3DGS models, decreasing the confidence level to 12%. Conversely, renders from test camera locations cause a drastic decrease in the confidence level of the system, though this varies across different object classes. The overall degradation in performance demonstrates that the proposed M-IFGSM attack effectively transfers to 3DGS models, though some specific cases, such as the couch dataset, indicate potential avenues for improving adversarial robustness.

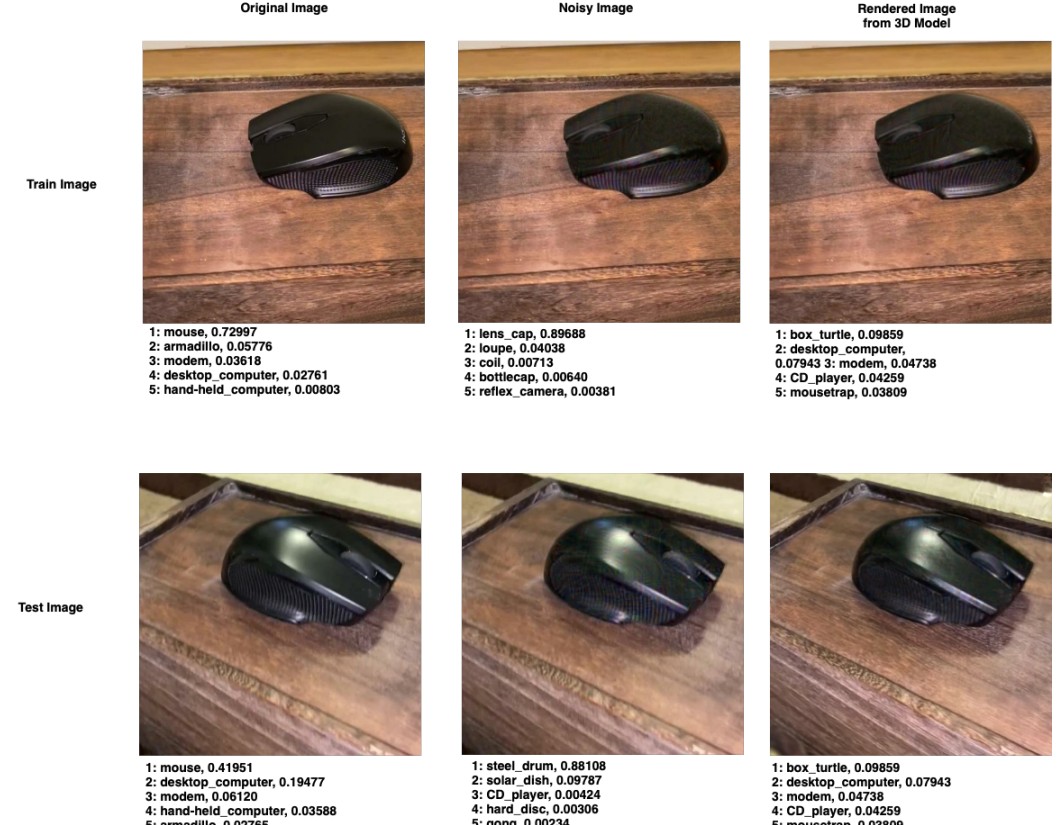

Figure 4: Adversarial noise effect on object detection. Sample images of the original image dataset (left top and bottom) and their attacked counterparts (middle top and bottom) and prediction from 3D model renders (right top and bottom). The bottom row is from a test image, indicating that the 3D model has not updated for this veiw

## 4    Conclusions

In this paper, we investigated adversarial attacks targeting vision-language models, specifically focusing on CLIP's object detection capabilities, and explored the transferability of these attacks to 3D models. Our study primarily centered on the application of the Masked Iterative Fast Gradient Sign Method (M-IFGSM), and we demonstrated how this adversarial noise could be effectively integrated into 3D Gaussian Splatting models. By employing the Common Objects 3D (CO3D) dataset, we conducted experiments on eight distinct object classes, creating noisy 3D models with adversarial noise from 35 images captured from different angles.

The significance of this work lies in its extension of adversarial attacks from 2D vision systems to 3D object detection models, which presents a previously unexplored area of research. Our findings indicate that, with the addition of segmentation, the adversarial noise designed for 2D models not only successfully transfers to the 3D domain but also leads to a substantial degradation in model performance. While the target class was consistently identified as the top-1 prediction in the original 3D models, under adversarial conditions, the model's top-1 accuracy sharply decreased. For the training images, the average top-1 accuracy fell from 95.4% to 12.5%, and for the test images, it dropped from 91.2% to 35.4%. This demonstrates that adversarial perturbations in a 2D context can profoundly impact the accuracy of 3D models, including those constructed from Gaussian Splatting techniques.

This work represents a novel contribution to the adversarial machine learning field by bridging the gap between 2D adversarial attacks and their effects in 3D environments. Additionally, our approach highlights the vulnerabilities of cutting-edge models like CLIP when tasked with multi-view object

recognition, pointing toward the necessity of building more robust models capable of withstanding adversarial manipulations across both 2D and 3D domains.

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
