# OpenReview forum: "Gaussian Splatting Under Attack: Investigating Adversarial Noise in 3D Objects"
_NeurIPS.cc/2024/Workshop/SafeGenAi — SafeGenAi Poster_

### Official Review · Reviewer_6GuR · 2024-10-09
**Comments**

**Rating:** 6
**Confidence:** 3

**Review:**

Summary:

(+) The paper presents a fresh approach by investigating adversarial attacks in the context of 3D Gaussian Splatting, which is a relatively underexplored area compared to traditional 2D adversarial attacks. This novel perspective contributes significantly to the field of adversarial machine learning, expanding the understanding of vulnerabilities in 3D vision models.

(+) The experiments conducted are comprehensive and yield intriguing results, such as the substantial drop in model accuracy and confidence levels due to the adversarial perturbations. These findings highlight critical vulnerabilities and suggest potential avenues for future research, making a compelling case for further exploration in this domain.

Minor suggestions:

The caption of table should be above the tabular.

Figures should be in higher resolution or vector graphic format.

---

### Official Review · Reviewer_P2v7 · 2024-10-09
**A well presented research work that shows novelty in adversarial attacks of 3D objects**

**Rating:** 9
**Confidence:** 3

**Review:**

Quality:
The work is presented well with detailed results of the experiment for attacking the effectiveness of Object classification with CLIP.

Clarity:
Overall, the work contains clearly formulated solution and concise presentation.
Though, there a few points that needs to be added:
1. It is unclear what specific model 𝑓 refers to in "Algorithm 1: Masked Iterative FGSM Adversarial Attack." However, based on the context of the algorithm and its application to adversarial attacks, it is likely that 𝑓 represents the CLIP model, which is used to compute the loss by classifying the input image with respect to its true label.
2. Clip_{min,max} is not clear what is being done here, I assume the images pixel intensity is clipped between valid RGB range of 0 to 255.

Originality:
A simple and novel formula being applied to the Adverserial attack.

Significance:
Since GS is hot area of research in spatial intelligence research, understanding of adversarial attack methods is significant in the future of where Gaussian Splatting is used in downstream tasks where computer vision algorithms are run.

Suggested points to add to the work:
1. Cover more diverse objects and datasets, including outdoor object like building structures.
2. Transferability of adversarial attacks: If model 𝑓, that is used to compute the gradient is a CLIP model, then, would the adverserially attacked 3D object mislead other pretrained Object classification vision models to predict the wrong class?

---

### Official Review · Reviewer_kqP3 · 2024-10-09
**Good quality paper with interesting idea. May be interesting to see testing/extension to other vision-language models.**

**Rating:** 7
**Confidence:** 4

**Review:**

Pros: The authors provide a novel contribution to extend an adversarial attack from 2d to 3d models. The attack is successful, showing that the average top-1 accuracy falls off significantly from its pre-attack state. There is also a number of interesting examples showing the different categorizations of perturbed images. The approach also allows for a targeted attack to be performed, improving over previous untargeted attacks which perturbed the whole image. The connection to safe generative AI is clear as it connects directly to describe current vulnerabilities to adversarial attacks.

Cons: Unclear if this methodology could be utilized on models other than CLIP, perhaps a future extension would be interesting here. The untargeted attack is also described as a current, but I do not see any experiments showing its use/efficacy? A comparison between the targeted and untargeted method would be useful.

Questions:
1) Could this work possibly be extended to other vision-language models?

2) You say you discuss real world implications of these attacks, but beyond showing the dramatic fall in accuracy, I don't see any other implication specifically highlighted?

---

### Official Review · Reviewer_ZYct · 2024-10-10
**Gaussian Splatting Under Attack: Investigating Adversarial Noise in 3D Objects**

**Rating:** 6
**Confidence:** 3

**Review:**

I like this work, It use a creative method to push the adversarial attack from 2-D to 3-D. here are the advantages and concerns:
Advantages:
1. The paper‘s writing is easy to understand.
2.The method used in the paper is inspiring and can inject adversarial noise imperceptibly。

Concerns:
1. The description of Figure 1 is not enough and it is difficult to understand the detailed process of the proposed method.
2. It is best to be able to make direct comparisons with related work.
3. Some minor details need to be modified, such as figure 4 appearing below the conclusion.